# Pyk2 Regulates MAMs and Mitochondrial Dynamics in Hippocampal Neurons

**DOI:** 10.3390/cells11050842

**Published:** 2022-03-01

**Authors:** Laura López-Molina, Joaquín Fernández-Irigoyen, Carmen Cifuentes-Díaz, Jordi Alberch, Jean-Antoine Girault, Enrique Santamaría, Silvia Ginés, Albert Giralt

**Affiliations:** 1Departament de Biomedicina, Facultat de Medicina, Institut de Neurociències, Universitat de Barcelona, 08036 Barcelona, Spain; laura.lopezmo12@ub.edu (L.L.-M.); alberch@ub.edu (J.A.); silviagines@ub.edu (S.G.); 2Institut d’Investigacions Biomèdiques August Pi i Sunyer (IDIBAPS), 08036 Barcelona, Spain; 3Centro de Investigación Biomédica en Red sobre Enfermedades Neurodegenerativas (CIBERNED), 28031 Madrid, Spain; 4Proteomics Platform, Navarrabiomed, Hospital Universitario de Navarra (HUN), Universidad Pública de Navarra UPNA, IdiSNA, 31008 Pamplona, Spain; joaquin.fernandez.irigoyen@navarra.es; 5Inserm UMR-S 1270, 75005 Paris, France; carmen.diaz@inserm.fr (C.C.-D.); jean-antoine.girault@inserm.fr (J.-A.G.); 6Sorbonne Université, 75005 Paris, France; 7Institut du Fer à Moulin, 75005 Paris, France; 8Production and Validation Centre of Advanced Therapies (Creatio), Faculty of Medicine and Health Science, University of Barcelona, 08036 Barcelona, Spain; 9Clinical Neuroproteomics Unit, Navarrabiomed, Hospital Universitario de Navarra (HUN), Universidad Pública de Navarra UPNA, IdiSNA, 31008 Pamplona, Spain; enrique.santamaria.martinez@navarra.es

**Keywords:** hippocampus, calcium, ER-mitochondria contact sites, neuron

## Abstract

Pyk2 is a non-receptor tyrosine kinase enriched in hippocampal neurons, which can be activated by calcium-dependent mechanisms. In neurons, Pyk2 is mostly localised in the cytosol and dendritic shafts but can translocate to spines and/or to the nucleus. Here, we explore the function of a new localisation of Pyk2 in mitochondria-associated membranes (MAMs), a subdomain of ER-mitochondria surface that acts as a signalling hub in calcium regulation. To test the role of Pyk2 in MAMs’ calcium transport, we used full Pyk2 knockout mice (Pyk2^−/−^) for in vivo and in vitro studies. Here we report that Pyk2^−/−^ hippocampal neurons present increased ER-mitochondrial contacts along with defective calcium homeostasis. We also show how the absence of Pyk2 modulates mitochondrial dynamics and morphology. Taken all together, our results point out that Pyk2 could be highly relevant in the modulation of ER-mitochondria calcium efflux, affecting in turn mitochondrial function.

## 1. Introduction

Proline-rich tyrosine kinase 2 (Pyk2) is a Ca^2+^-activated non-receptor tyrosine kinase belonging to the focal adhesion kinase (FAK) family [1]. Pyk2 is enriched in hippocampal adult neurons and plays an important role in neuronal plasticity and hippocampal-related memory [2]. In neurons, Pyk2 is mostly localised in the cytosol and dendritic shafts [3,4]. However, we and others have previously demonstrated that following neuronal activation, Pyk2 translocates to other subcellular compartments, such as the nucleus or the excitatory synapses [5,6]. Indeed, several studies have suggested Pyk2 as a key regulator in synaptic plasticity involved with both LTP and LTD [2,7,8]. In this line, we and others have shown that this role of Pyk2 in synaptic plasticity is compromised in different neurodegenerative disorders, such as Alzheimer’s disease and Huntington’s disease [2,7,8,9].

Besides this nuclear and synaptic localisation, following Ca^2+^ entry into the cell via the transient receptor potential (TRP) channel, Pyk2 can be phosphorylated and partially translocated to mitochondria [10,11]. So far, Pyk2 translocation to mitochondria has been described in cardiac cells, where Pyk2 can be activated and phosphorylate the mitochondrial calcium uniporter (MCU) [11,12,13] enhancing mitochondrial Ca^2+^ uptake and mitochondrial ROS production [12]. However, little is known about the molecular mechanism on how Pyk2 can regulate mitochondrial morphology and its consequent role in neuronal physiology.

Indeed, mitochondria are fundamental regulators in synapses since they can travel from soma to synaptic terminals to supply local ATP production and Ca^2+^ buffering [14]. To ensure this mitochondrial distribution and number, mitochondria morphology by means of fusion and fission events is critical and both processes must be coordinated [15]. Hence, alterations in mitochondrial morphology can hamper their transport within the cell and could result in synaptic dysfunction and Ca^2+^ mishandling [16]. Indeed, to regulate Ca^2+^ homeostasis, mitochondria interact with other subcellular compartments such as endoplasmic reticulum (ER), forming the mitochondria-associated membranes (MAMs), a hotspot for Ca^2+^ signalling. In this specialised region, calcium is transported from the ER through the IP3R and enters the mitochondria, crossing the channel VDAC1 (voltage dependent “anion” channel 1, which can be permeable to Ca^2+^). A proper establishment of ER-mitochondria contacts is crucial not only for Ca^2+^ homeostasis but also for the control of mitochondrial dynamics and morphology [17]. Indeed, defects in MAMs compromise neuronal functionality and survival [18,19] and several studies have shown either upregulation or disruption of ER-mitochondrial crosstalk in different neurological disorders such as Alzheimer’s disease (AD) [19], Parkinson’s disease (PD) [20,21], Huntington’s disease (HD) [22] and anxiety and depression [23].

In the current work, we aimed to study whether Pyk2 also localises in MAMs and its relevance in neuronal function. We observed that neuronal Pyk2 interacts with several molecules relevant for mitochondrial and endoplasmic reticulum physiology, while the absence of Pyk2 in hippocampal neurons increases the ER-mitochondria interaction leading to a disrupted Ca^2+^ buffering. Moreover, we show that Pyk2 modulates fission–fusion events. With all this evidence, we postulate Pyk2 as an essential player in the function of MAMs.

## 2. Materials and Methods

### 2.1. Animals

For this study, full knock-out mice for Pyk2 (Pyk2^−/−^) were used [24]. Animals were genotyped from a tail biopsy. The housing room was kept at 19–22 °C, under a 12:12 h light/dark cycle and animals had access to food and water ad libitum. All mice used in the present study were males of 5 months of age. All animal procedures were approved by local committees (Universitat de Barcelona, CEEA (10141) and Generalitat de Catalunya (DAAM 315/18)), in accordance with the European Communities Council Directive (86/609/EU).

### 2.2. Immnuoprecipitation

Hippocampal tissue was homogenised on ice-cold immunoprecipitation (IP) buffer (50 mM Tris, 150 mM NaCl, 2 mM EDTA, 1% Triton, pH 7.4, 1 mM PMSF, 10 μg/mL aprotinin, 1 μg/mL leupeptin). Four hundred µg of protein were incubated overnight at 4 °C under rotary agitation with 2 µg of Pyk2 antibody (Sigma, Burlington, MA, USA. #074M4755) or rabbit IgG (Jackson ImmunoResearch) as a negative control. The immune complexes were incubated precipitated overnight with the addition of sepharose beads (GE Healthcare, IL, Illinois, USA. 17-0618-01). Beads were washed once in IP buffer, IP/PBS (1:1) and PBS and collected by centrifugation at 3000 rpm 5 min. Immunocomplexes were heated 10 min at 100 °C and resolved on SDS-PAGE or resuspended in PBS for mass spectrometry analysis.

### 2.3. Mass Spectrometry

Immunoprecipitated protein extracts were diluted in Laemmli sample buffer and loaded into a 0.75 mm thick polyacrylamide gel with a 4% stacking gel casted over a 12.5% resolving gel. The run was stopped as soon as the front entered 3 mm into the resolving gel so that the whole proteome became concentrated in the stacking/resolving gel interface. Bands were stained with Coomassie Brilliant Blue, excised from the gel and protein enzymatic cleavage was carried out with trypsin (Promega, Madison, WI, USA; 1:20, *w*/*w*) at 37 °C for 16 h, as previously described [25]. Purification and concentration of peptides was performed using C18 Zip Tip Solid Phase Extraction (Millipore). Peptide mixtures were separated by reverse phase chromatography using an UltiMate 3000 UHLPC System (Thermo Scientific, Waltham, MA, USA) fitted with an Aurora packed emitter column (Ionopticks, 25 cm × 75 µm ID, 1.6 µm C18). Samples were first loaded for desalting and concentration into an Acclaim PepMap column (ThermoFisher, Waltham, MA, USA. 0.5 cm × 300 µm ID, 5 µm C18) packed with the same chemistry as the separating column. Mobile phases were 100% water 0.1% formic acid (FA) (buffer A) and 100% Acetonitrile 0.1% FA (buffer B). The column gradient was developed in a 120 min two step gradient from 5% B to 20% B in 90 min and 20% B to 32% B in 30 min. The column was equilibrated in 95% B for 10 min and 5% B for 20 min. During all processes, the precolumn was in line with the column and the flow maintained all along the gradient at 300 nL/min. The column temperature was maintained at 40 °C using an integrated column oven (PRSO-V2, Sonation, Biberach, Germany) and interfaced online with the Orbitrap Exploris 480 MS. Spray voltage were set to 2 kV, funnel RF level at 40, and heated capillary temperature at 300 °C. For the DDA experiments full MS resolutions were set to 1,200,000 at *m*/*z* 200 and the full MS AGC target was set to Standard with an IT mode Auto. The mass range was set to 375–1500. The AGC target value for fragment spectra was set to Standard with a resolution of 15,000 and 3 s for cycle time. Intensity threshold was kept at 8E3. Isolation width was set at 1.4 *m*/*z*. Normalised collision energy was set at 30%. All data were acquired in centroid mode using positive polarity, and peptide match was set to off and isotope exclusion was on.

Raw files were processed with MaxQuant [26] v1.6.17.0 using the integrated Andromeda Search engine [27]. All data were searched against a target/decoy version of the mouse Uniprot Reference Proteome without isoforms (55,366 entries) from the March 2021 release. First search peptide tolerance was set to 20 ppm, main search peptide tolerance was set to 4.5 ppm. Fragment mass tolerance was set to 20 ppm. Trypsin was specified as enzyme, cleaving after all lysine and arginine residues and allowing up to two missed cleavages. Carbamidomethylation of cysteine was specified as fixed modification and peptide N-terminal acetylation, oxidation of methionine, deamidation of asparagine and glutamine and pyro-glutamate formation from glutamine and glutamate were considered variable modifications with a total of two variable modifications per peptide. “Maximum peptide mass” was set to 7500 Da, the “modified peptide minimum score” and “unmodified peptide minimum score” were set to 25 and everything else was set to the default values, including the false discovery rate limit of 1% on both the peptide and protein levels. The Perseus software (version 1.6.14.0) was used for statistical analysis and data visualization.

### 2.4. Electron Microscopy

Mice were transcardially perfused with a solution containing 4% paraformaldehyde and 0.1% glutaraldehyde (both weight/vol) made up in 0.1 M phosphate buffer (PB), pH 7.4. After perfusion, the brains were removed from the skull, and immersed in the same fixative 12 h 4 °C. Tissue blocks containing the hippocampus were dissected, washed with PB and cut with a vibratome to obtain samples of 1 mm^3^. Hippocampal neuronal cultures after 21 days in vitro were treated with 40 µM glutamate or vehicle and then fixed 30 min with the same fixative solution used for the tissues and washed with PB. Samples from tissues or cell cultures were post-fixed with 2% osmium tetroxide in PB 0.1 M for 20 min. They were dehydrated in a series of ethanol and flat embedded in epoxy resin (EPON 812 Polysciences, Warrington, PA, USA). After polymerisation, blocks from the CA1 region or blocks from cell cultures were cut at 70 nm thickness using an ultramicrotome (Ultracut E Leica, Wetzlar, Germany). Sections were cut with a diamond knife, picked up on formvar-coated 200 mesh nickel grids. They were then immunostained by indirect immunolabeling using protein gold as immunomarker. Protein A-gold probes (20 nm) were obtained from CMC Utrecht (Utrecht, The Netherlands). After immunolabeling, the sections were double stained with uranyl acetate and lead citrate, prior to observation with a Philips (CM-100) electron microscope. Digital images were obtained with a CCD camera (Gatan Orius. Las Positas Blvd, Pleasanton, CA, USA). EPON embedded sections were ultrathin-sectioned and reacted with anti-Pyk2 (Sigma, Burlington, MA, USA. #P3902) antibody revealed with protein A coupled to 10 nm gold particles. For particle counting in animal tissue and in cell cultures, the number of individual gold particles (10 nm) localised in morphometrically determined areas was hand counted and the labelling density was calculated as the number of gold particles per µm and then we relativised the values to a percentage. Each mitochondrial contact closely associated with ER was counted. We considered contact of the mitochondria to ER to be if their distance was less than 30 nm, as previously described [28]. We counted the number of these contacts per cell and fifty mitochondria from six cells per condition were analysed.

### 2.5. Mitochondrial Isolation

Hippocampal samples were removed and homogenised in buffer A (20 mM HEPES, 2 mM EDTA, 1.5 mM MgCl_2_, 10 mM KCL, pH 7.5, 1 mM PMSF, 10 μg/mL aprotinin, 1 μg/mL leupeptin, 2 mM sodium orthovanadate) using two steps of mechanical disintegration. The homogenates were centrifuged at 500× *g* for 5 min, with the resulting pellet (P1) discarded and supernatant (S1) being centrifuged at 13,000× *g* for 20 min. The resulting second supernatant (S2) was discarded, and the pellet (P2) was resuspended with Buffer A (250 mM Sucrose).

### 2.6. Immunoblot Analysis

Animals were sacrificed by cervical dislocation and the dorsal hippocampus was rapidly dissected on ice and stored at −80 °C until use. Tissue was lysed by sonication in 250 µL of lysis buffer (PBS, 10 mL L^−1^ Nonidet P-40, 1 mM PMSF, 10 mg L^−1^ aprotinin, 1 mg L^−1^ leupeptin and 2 mg L^−1^ sodium orthovanadate). After lysis, samples were centrifuged at 15,000× *g* for 20 min. Supernatant proteins (15 mg) from hippocampal brain regions extracts were loaded in SDS–PAGE and transferred to nitrocellulose membranes (GE Healthcare, Chicago, IL, USA). Membranes were blocked in TBS-T (150 mM NaCl, 20 mM Tris-HCl, pH 7.5, 0.5 mL L^−1^ Tween 20) with 50 g L^−1^ non-fat dry milk and 5 g L^−1^ BSA. Membranes were probed overnight at 4 °C by shaking with the following primary antibodies (all diluted 1:1000 with some exceptions): rabbit polyclonal antibodies: Pyk2 (Sigma 074M4755), Lamin B (Santa Cruz sc-6217), CoxV (Invitrogen A21347), IP3R3 (1:500, BD Bioscience 610312), VDAC1 (Abcam, ab15895), TOM20 (Abcam, ab56783), Drp1 (BD Bioscience 611113), Opa-1 (1:8000, BD Bioscience 612607), Mitofusin-2 (1:500, Abcam ab56889). Then, membranes were incubated with anti-rabbit or anti-mouse horseradish peroxidase-conjugated secondary antibody (1:30,000; Promega W4021 or W4011). Secondary antibody binding was detected by Luminol reagent (Santa Cruz sc-2048). For loading control, a mouse monoclonal antibody for a-tubulin (1:30,000; Sigma 083M4847V,) or a-actin (1:30,000; Sigma A3854) was used.

### 2.7. Primary Cultures of Hippocampal Neurons

Hippocampal neurons were obtained from E17.5 Pyk2^+/+^ and Pyk2^−/−^ mice. The hippocampus was dissected and mechanically dissociated with a fir-polished Pasteur pipette. Cells were seeded at a density of 50,000 or 100,000 neurons for immunocytochemistry and transfections, respectively, onto 12 mm coverslips placed in 24-well plates or at a density of 80–90,000 neurons onto 25 mm coverslips placed in 6-well plates for calcium analysis. Plates were previously precoated with 0.1 mg/mL poly-D-lysine (Sigma Chemical Co., St. Louis, MO, USA) and neurons were cultured in Neurobasal medium (Gibco-BRL, Renfrewshire, UK) supplemented with 1% Glutamax and 2% B27 (Gibco-BRL). Cultures were maintained at 37 °C in a humidified atmosphere containing 5% CO_2_. All experiments with neuronal cultures were analysed at DIV21.

### 2.8. Immunocytochemistry and Confocal Imaging

Primary neurons were fixed at DIV21 with 4% paraformaldehyde solution in PBS for 10 min and blocked in PBS-0.1 M glycine for 10 min. Then, cells were permeabilised in PBS-0.1% saponine 10 min, blocked with PBS-Normal Horse Serum 15% 30 min and incubated overnight at 4 °C in the presence of the following primary antibodies: rabbit TOM-20 (1:250, ProteinTech 11802-1-AP) and mouse MAP2 (1:500, Sigma-Aldrich M1406). Fluorescent secondary antibodies: AlexaFluor 488 goat anti-rabbit (1:100), Cy3 goat anti-mouse (1:100), and/or Cy3 goat anti-mouse (1:100; all three from Jackson ImmunoResearch, West Grove, PA, USA) were incubated for 1 h at RT. Nuclei were stained with DAPI-Fluoromount (SouthernBiotech, Birmingham, AL, USA). Immunofluorescence was analysed using a Leica Confocal SP5-II confocal microscope (Leica Microsystems CMS GmbH, Mannheim, Germany). Images were taken using a HCX PL APO lambda blue 63.0 × 1.40 OIL objective with a standard pinhole (1 AU), at 1024 × 1024-pixel resolution, 0.4 m thick and 3.0 digital zoom.

### 2.9. Proximity Ligation Assay

ER-mitochondria contact sites were analysed performing a proximity ligation assay (Duolink, DUO92101–1KT) following manufacturer’s instructions. After fixation and blocking, neurons were incubated with MAP2 to stain cell area. Then, antibodies IP3R3 (1:500, Merck AB9076) and VDAC1 (1:500, Abcam ab15895) were incubated overnight to label ER and mitochondria, respectively. PLA probes anti-mouse PLUS and anti-rabbit MINUS were incubated for 1 h at 37 °C and hybridisation was amplified by rolling circle amplification. Nuclei were stained with DAPI using mounting media provided in the kit. Confocal images were taken using 3.0 digital zoom and stacks of 0.4 µm. Fluorescent dots were counted using ImageJ and number of particles were relativised to cell area.

### 2.10. Calcium Imaging in Neuronal Cultures

To study calcium handling capacity from ER and mitochondria, calcium imaging was performed in hippocampal neuronal cultures. Pyk2^+/+^ and Pyk2^−/−^ neurons at DIV21 were washed with Krebs buffer and loaded with 5 µM Fluo-4 (Invitrogen F-14201) for 30 min and then with 20 nM TMRM (Invitrogen T668) for 20 min at RT to detect intracellular Ca^2+^ changes (Cai^2+^) and mitochondrial membrane potential (ΔΨm), respectively. Coverslips were assembled in a chamber filled with 500µl of Krebs buffer on the stage of confocal microscope with an incubator system that controls temperature and CO_2_. After 50 s, 0.5 µM thapsigargin (TG, Sigma-Aldrich T9033) was injected to inhibit SERCA ATPase and block Ca^2+^ storage at the ER. Then, 2 µM FCCP (Sigma-Aldrich C2920) were added at min 7.5 to depolarise mitochondrial membrane and inducing Ca^2+^ release to the cytosol. Images were captured every 2.5 s throughout 750 s of experiment. Fluorescence intensity was quantified using ImageJ software and results are expressed as absolute values of intensity.

### 2.11. Mitochondrial Morphology

For electron microscopy, images were obtained from mouse brain sections. The number of mitochondria, aspect ratio (AR) and form factor (FF) were quantified by counting the total number of visible mitochondria present in a fixed size surface of the *stratum radiatum* from the CA1 region. AR and FF were determined manually tracing individual mitochondria using ImageJ software (NIH, Bethesda, MD, USA). For cell immunocytochemistry, mitochondrial morphology was quantified as previously described [22]. Images were processed with ImageJ software using a convolve filter and automatically given a threshold to obtain binary images. Using the plug-in “Analyse particles”, individual mitochondria were measured to calculate the number of mitochondria per micron, AR (length-to-width ratio), and FF (Pm2/4 Am), where Pm is the perimeter and Am is the area of mitochondrion. AR values of 1 mean a perfect circle, whereas higher values correspond to more elongated mitochondria. FF values of 1 represent unbranched mitochondrion and higher FF values indicate a more complex mitochondria network.

### 2.12. Cell Transfection

Pyk2^+/+^ and Pyk2^−/−^ hippocampal neurons were transfected at DIV 18 using transfectin (Bio-Rad) following the manufacturer’s instructions and incubated for 72 h. Cells were transfected with several Pyk2 constructs previously described [2,29]: GFP (control), GFP-Pyk2, GFP-Pyk2^-DFAT^ (FAT domain and the third proline-rich motif were deleted from Pyk2), GFP-Pyk2^YF^ (Pyk2 with a point mutation of the autophosphorylated tyrosine-402) and GFP-Pyk2^RRST^ (Pyk2 with four point mutations in nuclear transport and location motifs). GFP was fused to the N-terminus of Pyk2.

### 2.13. Statistical Analysis

Statistical analyses were carried out using the GraphPad Prism 8.0 software (GraphPad, San Diego, CA. USA). All experiments were blind coded to the experimenter. Two-tailed Student’s *t*-test (95% confidence), one-way ANOVA or two-way ANOVA, with Tukey’s or Bonferroni’s post hoc multiple comparison test was performed as required if distributions were normal. In case of not normal distributions Mann–Whitney *t*-test or the Dunn’s test were used. A *p* value <0.05 was considered significant.

## 3. Results

### 3.1. Pyk2 Is Localised in Mitochondria and MAMs and Interacts with Specific Partners

The physiological and molecular role of Pyk2 in the brain is not yet completely understood. We considered that the elucidation of the Pyk2 interactome could help to decipher its direct association with specific cellular processes in the brain. For that reason, Pyk2 was immunoprecipitated from the hippocampus of Pyk2^+/+^ mice. Pyk2^−/−^ hippocampal material as well as IgG irrelevant antibodies were included as negative controls to detect non-specific–associated proteins. First, hippocampal Pyk2 immunoprecipitation was confirmed using Western blot (Figure 1A). Subsequent mass-spectrometry analysis was performed to identify Pyk2-associated proteins (Figure 1B and Appendix A). Data were curated by excluding non-specific–associated proteins detected in both Pyk2^−/−^ and IgG immunoprecipitations. After restrictive curation, 52 proteins were considered as co-immunoprecipitating with Pyk2. This Pyk2 interactome was functionally analysed using Metascape [30]. As shown in Figure 1C, multiple functional categories were significantly enriched (Appendix A). “NMDA receptor activation events” (LogP: −13.95) were represented by 34 Pyk2 associated-proteins (CALM3, CAMK2D, PRKAR2A, TUBA4A, TUBA1B, TUBB3, TUBB4A, TUBB4B, TBAL3, TCP1, GNB1, PRKCG, YWHAQ, DYNC1H1, HBA1, RPL8, RPL18, RPL23A, RPL27A, SOD2, PRDX5, ATP5F1C, SYT7, MBP, PPP3CB, UBA1, CYFIP2, ARHGAP26, ATP2B1, SRSF3, MAP1A, MAP1B, MAP4, KHDRBS1). Fifteen Pyk2 interactors were associated with “Synaptic plasticity regulation” (LogP = −8.84, MAP1A, MAP1B, PPP3CB, PRKCG, TNR, SYT7, SYNGR1, CALM3, MAP4, TUBB3, CYFIP2, PLEC, PXN, GPI, SOD2) whereas 24 associated proteins were directly related with “calcium regulation” (LogP = −8.16, ATP2B1, CALM3, CAMK2D, GNB1, PRKAR2A, PRKCG, YWHAQ, PPP3CB, PFKL, PFKM, PXN, CTBP1, SYT7, ENO1, RAB3C, TNR, SOD2, MAP1B, CYFIP2, TUBA4A, HBA1, PLEC, ATP5F1C, MBP). Interestingly, biofunctions such as “generation of precursor metabolites and energy” (LogP = −6.56, 17 Pyk2-associated proteins; ATP5F1C, ENO1, GPI, OXCT1, PFKL, PFKM, PLEC, SOD2, CAMK2D, PRKCG, CALM3, PRDX5, TUBA4A, DYNC1H1, SYNGR1, TUBB4B, CBR1) and “mitochondrial biogenesis” (LogP = −3.25, 6 Pyk2-associated proteins; ATP5F1C, CALM3, SOD2, PFKM, PLEC, CBR1) were also significantly over-represented in our Pyk2-interactome data. In summary, the results confirmed the main function of Pyk2 as a downstream regulator of NMDA receptor activity and synaptic plasticity, but they also highlighted a novel role of Pyk2 in the regulation of calcium homeostasis and mitochondrial function in neurons.

### 3.2. The Lack of Pyk2 Increases ER-Mitochondria Contact Sites

We next explored Pyk2 localisation in the mitochondria of the CA1 pyramidal neurons from Pyk2^+/+^ mice by electron microscopy. Pyk2-positive immunogold particles were found within mitochondria and its interactions with ER (Figure 2A). To test method specificity of the immunogold staining procedure, the primary antibody was omitted. Under these conditions, no selective labelling was observed (Figure 2B). Moreover, we validated these results performing a subcellular fractionation and isolating mitochondria. Pyk2 was detected both in total lysate and the different cellular compartments, including nucleus, cytosol and mitochondria (Figure 2C).

Since we saw that Pyk2 was present at MAMs, we wondered whether this protein could affect the establishment of contact sites between the two implicated organelles. First, we evaluated levels of MAM-resident proteins in hippocampal total lysates. Pyk2^−/−^ showed lower IP3R3 and higher VDAC1 protein levels compared to Pyk2^+/+^, suggesting structural alterations of MAMs (Figure 3A). To further examine these organelles associations, we next measured ER–mitochondria interactions both in vivo and in vitro neurons of Pyk2^+/+^ and Pyk2^−/−^ mice. Hippocampal sections were processed for electron microscopy and analysed. Quantification of the number of MAMs revealed an increased number of contact sites in Pyk2^−/−^ neurons compared to Pyk2^+/+^ (Figure 3B). We also measured ER–mitochondria association in hippocampal neuronal cultures using a proximity ligation assay (PLA) labelling IP3R3 and VDAC1 as probes for ER and mitochondria, respectively. In line with the in vivo results, Pyk2^−/−^ neurons presented an increase in the number of interactions between organelles compared to Pyk2^+/+^ neurons (Figure 3C).

### 3.3. Pyk2 Is Involved in Calcium Homeostasis in Hippocampal Neuronal Cultures

MAMs are specialised regions highly implicated in Ca^2+^ homeostasis. Given the observed changes in the number of contact sites in Pyk2^−/−^ neurons, we aimed to study if the absence of Pyk2 could also be affecting Ca^2+^ intracellular levels. Therefore, levels of intracellular Ca^2+^ (Cai^2+^) and mitochondrial membrane potential (Δψm) were labelled using the dyes Fluo4 and TMRM, respectively, in hippocampal Pyk2^+/+^ and Pyk2^−/−^ neurons (Figure 4A and Appendix A). In basal conditions, no differences in fluorescence intensity were detected between genotypes. Then, thapsigargin (TG) was injected to inhibit SERCA (sarco-endoplasmic reticulum Ca^2+^-ATPases) activity and to deplete the Ca^2+^ store in the ER. As expected, after the treatment, Pyk2^+/+^ cells significantly increased Cai^2+^ levels and lowered Δψm. On the contrary, in Pyk2^−/−^ neurons thapsigargin did not induce a significant increase in Cai2+ levels, either compared to Pyk2^+/+^ or to Pyk2^−/−^ in basal conditions. Consistent with these results, Δψm remained unchanged. Next, cells were incubated with FCCP, an oxidative phosphorylation uncoupler, to induce maximal mitochondria depolarisation. Right after this stress, Cai^2+^ levels raised in Pyk2^+/+^ neurons and, to a lesser extent, in Pyk2^−/−^ cells. Accordingly, mitochondria depolarisation was observed in Pyk2^+/+^ neurons, manifested as a significant decrease in TMRM fluorescence (Figure 4B,C). Since the mitochondrial calcium uptake strongly depends on the cytoplasmic calcium concentration [31,32], less reduction in the Δψm of Pyk2^−/−^ neurons was detected. This effect could be attributable to the lower response to thapsigargin observed in Pyk2^−/−^ cells. Altogether this experiment demonstrates that ER depletion in the absence of Pyk2 leads to smaller cytoplasmic calcium elevation, which could be a sign of altered calcium capacity of ER in this condition.

### 3.4. Pyk2 Modulates Mitochondrial Morphology In Vivo and In Vitro

One of the many functions of MAMs is the regulation of mitochondrial dynamics [17]. Considering that the absence of Pyk2 triggers defects in ER–mitochondrial interaction, we wondered whether this could influence mitochondrial dynamics. Thus, we evaluated mitochondrial morphology in CA1 pyramidal neurons in hippocampal sections from both Pyk2^+/+^ and Pyk2^−/−^ mice (Figure 5A). Pyk2^−/−^ mice showed a higher mitochondria density (Figure 5B), along with a reduction in mitochondrial length (Figure 5C) and mitochondrial branching (Figure 5D). Furthermore, Pyk2^−/−^ mice showed an increase in hippocampal TOM20 levels (Figure 5E,F), corroborating the increase in the number of mitochondria. Additionally, Drp1 protein levels were increased in Pyk2^−/−^ mice (Figure 5E,G) whereas both Opa1 and Mitofusin-2 protein levels presented no differences between genotypes (Figure 5H–J), suggesting that the excessive mitochondrial fragmentation is due to alterations in fission but not fusion processes.

We then assessed mitochondrial morphology in Pyk2^−/−^ hippocampal primary neurons. Cells were labelled with TOM20 and MAP2 and mitochondrial morphometric analysis was performed (Figure 6A). Pyk2^−/−^ neurons had an increased number of mitochondria, together with lower values of AR and FF, indicating mitochondrial fragmentation and validating previous in vivo results (Figure 6B–D). These results confirmed the data obtained in vivo.

### 3.5. Nuclear Import/Export Domains of Pyk2 Control Dynamics of Hippocampal Mitochondria

To explore the molecular mechanisms underlying mitochondrial regulation by Pyk2, we transfected Pyk2^−/−^ hippocampal primary neurons with several truncated forms of GFP-Pyk2: DFAT, YF, and RRST (Figure 7A). Mutated constructs were compared with GFP transfected cells both from Pyk2^+/+^ and Pyk2^−/−^ cultures. As expected, restoring Pyk2 levels in Pyk2^−/−^ cultures rescued mitochondria numbers in the hippocampal neurons (Figure 7B,C). Similarly, Pyk2^−/−^ neurons transduced with DFAT and YF constructs also rescued the number of mitochondria. Interestingly, the RRST construct was unable to rescue this parameter displaying the same phenotype as the one in Pyk2^−/−^ neurons only transfected with GFP (Figure 7B,C). These results suggest that the corresponding motifs in the Pyk2 protein (RR_184,185_, and/or S_747_–T_749_) are directly involved in the role of Pyk2 in mitochondria, for example in the binding of specific partners, or that the effects of Pyk2 on mitochondria indirectly requires the nuclear translocation of Pyk2 that is prevented in the RRST mutant [2,5]. In an attempt to assess this possible translocation of Pyk2 into the mitochondria, wild-type hippocampal neurons were treated with 40 µM glutamate for 15 min. Neurons were immunolabeled with Pyk2 and observed by electron microscopy to analyse particle distribution. Interestingly, after glutamate treatment, density of Pyk2 particles in mitochondria increased, supporting the idea of a translocation of Pyk2 into the mitochondria upon neuronal activation (Figure 7D,E).

## 4. Discussion

Pyk2 is considered to be a scaffolding protein with numerous protein interactions (reviewed in [33]). In this work, we have partially defined the Pyk2 hippocampal interactome using immunoprecipitation coupled to mass-spectrometry. Most of Pyk2-associated proteins are directly involved in NMDA function, calcium regulation and mitochondrial homeostasis. Among all these protein partners, Pyk2 has been described to interact with some MAMs proteins as Grp75 and Grp78/BiP in the glioblastoma cell line [34]. In ER-mitochondria contacts, Grp75 strengthens the IP3R3-VDAC1 complex whereas Grp78, together with SIGR1, stabilises IP3R3 and boosts Ca^2+^ signalling [35,36]. In line with these results, we found that the absence of Pyk2 promotes alterations in the protein levels of IP3R3 and VDAC1 accompanied with an increase in ER-mitochondria apposition in hippocampal neurons both in in vivo and in vitro murine models, suggesting Pyk2 as an important protein in the formation of MAMs in a physiological context. Conversely, the upregulated function of MAM together with increased levels of resident proteins has been described in AD mouse brain and cell line and human fibroblasts [19,37], as well as in PD in in vitro models [38].

Considering that MAMs are highly implicated in calcium homeostasis and given the structural alterations described in Pyk2^−/−^ mice, we postulate that Pyk2 is an important player in the calcium homeostasis regulated by ER and mitochondria. Our results suggested that Pyk2 depletion could hamper the calcium buffering capacity of ER. In line with this outcome, previous work in endothelial cells has reported that Pyk2 phosphorylates STIM1, a calcium sensor in the ER, thus modulating Ca^2+^ store depletion in the ER [39,40]. Since the methodology used in this work for calcium imaging only traced cytoplasmic calcium, we cannot conclude that Pyk2 directly modulates calcium transfer from ER to mitochondria. However, the observed loss of contact sites between organelles and alterations of MAMs resident proteins involved in calcium efflux suggests that calcium homeostasis can be affected in the absence of Pyk2 due to altered functional crosstalk of these microdomains. Hence, further investigation would be necessary to unravel the role of Pyk2 in ER-mitochondria calcium efflux.

Although the molecular mechanisms by which Pyk2 regulates mitochondrial function are scarcely understood, it has been proposed that Pyk2/MCU pathway modulates the mitochondrial calcium uptake [12,41]. Thus, in cardiomyocytes, adrenergic stimulation induces Pyk2 translocation from cytosol to mitochondrial matrix where it directly phosphorylates the mitochondrial calcium uniporter (MCU), facilitating Ca^2+^ entry into the mitochondria [12]. Other studies in cardiac and neuroblastoma cells have shown that Ca^2+^ entry via TRPM2 activates Pyk2/MCU signalling and modulates mitochondrial function and cell survival [10,11].

MAMs regulate not only calcium buffering but also mitochondrial dynamics [17]. Mitochondrial dynamics are essential to control mitochondrial morphology and distribution within the cell [14] and need to be balanced for a proper cellular function. In the present work, we show that Pyk2 suppression triggers an exacerbated mitochondrial fission but does not alter fusion events. Moreover, Pyk2^−/−^ hippocampal neurons present augmented numbers of mitochondria and a disrupted mitochondrial network. This mitochondrial fragmentation could compromise neuronal function and viability. Other studies have reported that ablation of fusion or fission proteins promotes an oxidative stress response, neuroinflammation and neuronal death in the hippocampus [42,43,44].

Nevertheless, although we describe here a new function of Pyk2 in mitochondrial dynamics, the underlying mechanism remains unclear. We tried to address how Pyk2 can control the changes in hippocampal mitochondria density and morphology. We showed that upon neuronal activation Pyk2 is capable of translocating to the neuronal mitochondria. Supporting this result, Pyk2 activation can induce its own translocation to the mitochondria in cell lines and trigger production of mitochondrial ROS [12]. Searching for the main Pyk2 domain in the regulation of mitochondria biogenesis we notably observed that, from the tested domains, the RRST was the most important. The latter result indicates that Pyk2 domains of nuclear location and transport are essential for the regulation of mitochondrial dynamics. These findings could indicate two potential independent mechanisms: on the one hand, Pyk2 needs to translocate into the mitochondria and, once there, it regulates undetermined physiological molecular pathways. Indeed, MAMs’ chaperone, HSPA8, is responsible for protein imports into the mitochondria [45] and it has been described as interacting with Pyk2 [34]. On the other hand, Pyk2 nuclear import motifs are necessary to translocate Pyk2 to the nucleus as previously shown [3,5] to regulate the transcription of genes [46] related with the function of the mitochondria. Supporting the later hypothesis, it is well known that the mitochondria and the nucleus are coordinated in several physiological conditions [47,48]. In this context, Pyk2 could be a molecular bridge of such a process. However, future research should work in that direction to disentangle the exact mechanism.

Taken together, our results identify Pyk2 as a key molecule in the calcium regulation at ER-mitochondria contact sites. Such calcium regulation is essential for the expression of several forms of synaptic plasticity. Synaptic plasticity is a complex process modulated upon neuronal activity and implicates functional and structural changes in neurotransmitters’ release, cytoskeleton reorganisation, gene transcription and protein synthesis [49]. To complete all these cellular and molecular modifications, synapses require a tight calcium modulation and high energetic demands. Thus, mitochondria arise as a fundamental regulator in synapses since they can travel from soma to synaptic terminals to supply local ATP production and Ca^2+^ buffering [14]. To ensure mitochondrial distribution and numbers, fusion and fission events must be coordinated [15]. Alterations in mitochondrial morphology can hamper their transport within the cell and could result in synaptic dysfunction and Ca^2+^ mishandling [16]. Accordingly, it is widely accepted that calcium mishandling is a hallmark of neurological diseases. In this line, Pyk2 has been strongly implicated in neurodegenerative and psychiatric disorders as HD [2], chronic stress [50] and AD [8,51]. Indeed, the encoding gene *PTK2B* has been strongly associated as a genetic risk factor for late-onset AD [52]. Hence, we conclude that further investigation on Pyk2’s functions in physiological conditions implicating ER-mitochondria contact sites will contribute to a better understanding on the pathophysiology of several brain diseases.

## Figures and Tables

**Figure 1 cells-11-00842-f001:**
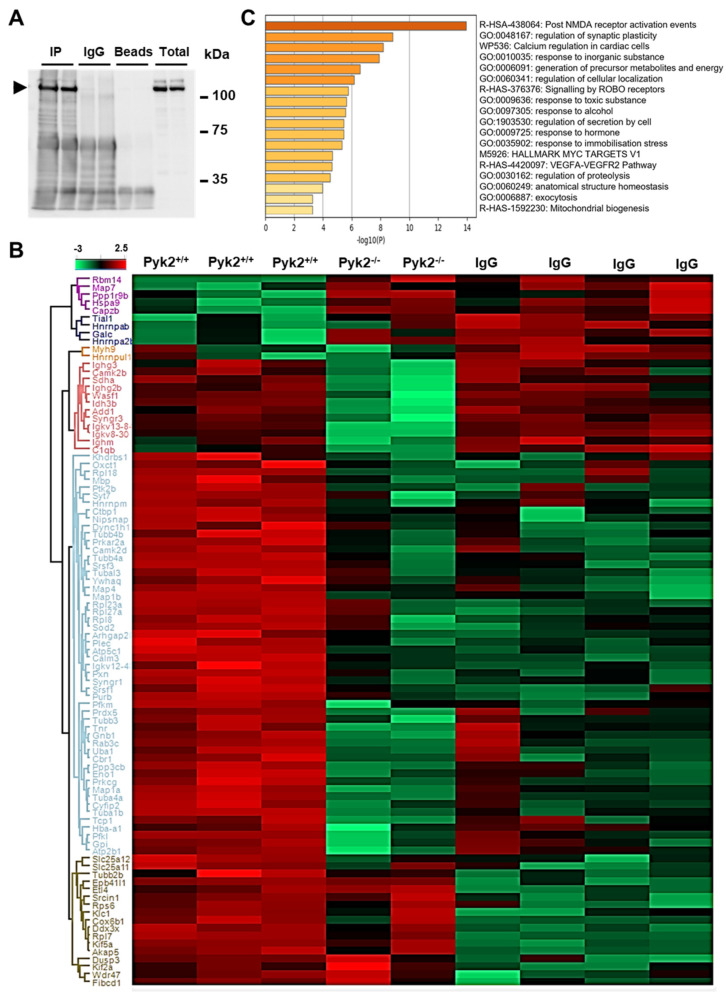
Identification of Pyk2 molecular interactors in the hippocampus. (**A**) Immunoprecipitation (IP) assay was performed in protein extracts from adult (4-month-old) wild-type hippocampal tissue. IgG antibodies were used as a control of the assay and total lysates were added as a control of the antibody immunoreactivity. The arrowhead depicts Pyk2; (**B**) Heatmap representation showing both clustering and the intensity for the immunoprecipitated proteins in each biological condition; (**C**) Statistically enriched biofunctions from Pyk2-associated proteins (from light blue cluster in Figure 1B; see Appendix A for details).

**Figure 2 cells-11-00842-f002:**
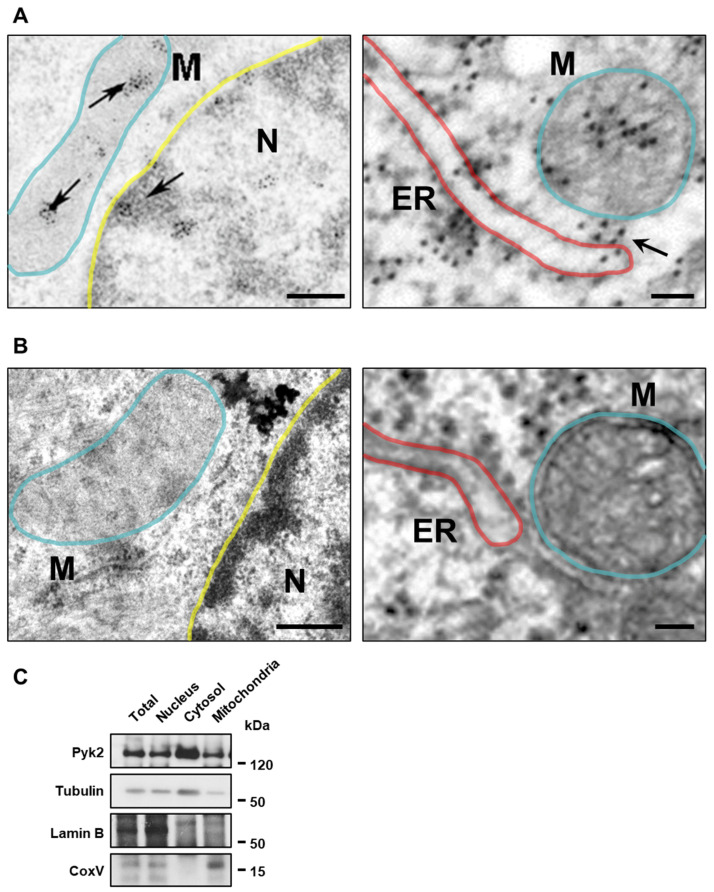
Characterisation of Pyk2 localisation in mitochondria and MAMs. (**A**) Electron microscopy images from hippocampal pyramidal neurons with Pyk2-immunogold labelling shows how Pyk2 is localised in the inner part of mitochondria (M), inside nucleus (N) and in the ER-mitochondria contact sites. Scale bars: left panel 0.4 µm, right panel 0.06 µm; (**B**) Negative control without Pyk2 antibody in hippocampal sections of Pyk2^+/+^ mice. Scale bars: left panel 0.4 µm, right panel 0.06 µm. In (**A**,**B**), blue lines delimit mitochondria, red lines the ER and yellow lines the nucleus; (**C**) Immunoblotting for Pyk2, Tubulin, Lamin B and CoxV in different subcellular fractions of a representative sample from mitochondrial isolation of hippocampus of Pyk2^+/+^ mouse.

**Figure 3 cells-11-00842-f003:**
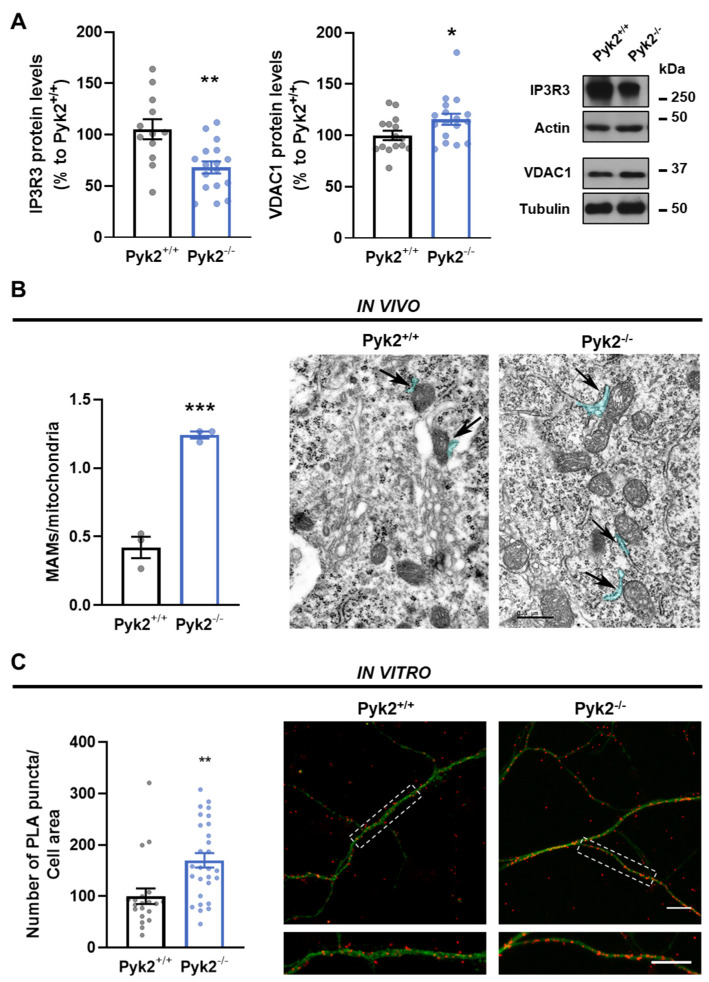
Pyk2 ablation induces increased number of ER–mitochondria contact sites in vivo and in vitro. (**A**) Densitometry quantification and representative immunoblotting of IP3R3 (Student *t*-test, *t* = 3.457, df = 27; * *p* < 0.05) and VDAC1 (Mann–Whitney test, A, B = 189, 339, U = 69; * *p* < 0.05) in total lysates of hippocampus from Pyk2^+/+^ and Pyk2^−/−^ mice. Actin or tubulin were used as loading control. Molecular weight markers positions are indicated in kDa in the right panel. Each point represents an animal; (**B**) Quantification of MAMs in vivo in hippocampal slices from mice (Student *t*-test, *t* = 9.936, df = 4; *p* < 0.001) and representative images of electron microscopy. Black arrows indicate MAMs and blue area delineate the analysed region. Scale bar, 0.5 µm; (**C**) MAMs were quantified in vitro by proximity ligation assay measuring VDAC1-IP3R3 interaction in neurites of hippocampal primary neurons. In (**C**) left panel, quantification shows more MAMs (Student *t*-test, *t* = 3.307, df = 46; *p* < 0.01) in Pyk2^−/−^ primary neurons. In (**C**) right panel, representative confocal images with interactions between the two targeted proteins in red and anti-MAP2 in green. Scale bar, 10 microns. Data are means ± SEM. Student *t*-test *p* values are: ** *p* < 0.01, *** *p* < 0.001 vs. Pyk2^+/+^. In (**A**), n = 12–19 animals/group; in (**B**), 50 mitochondria of 6 neurons, from 3 different animals.; in (**C**), n = 20–28 neurons/group from 4 different cultures.

**Figure 4 cells-11-00842-f004:**
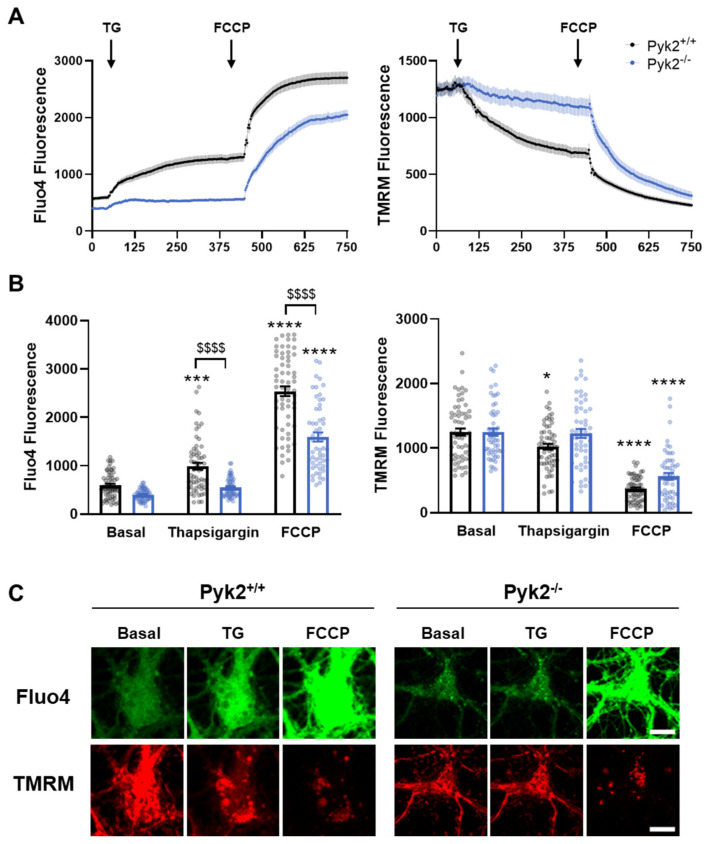
Regulation of cytosolic calcium levels is affected in hippocampal neurons devoid of Pyk2. Pyk2^+/+^ and Pyk2^−/−^ primary neurons were loaded with Fluo4 (5 μM) and TMRM (20 nM) to label intracellular calcium and mitochondrial membrane potential, respectively. Neurons were stimulated with thapsigargin (TG, 0.5 μM) at 50 s and with FCCP (2 μM) at 7.5 min. (**A**) Fluo4 and TMRM fluorescence traces from Pyk2^+/+^ (black) and Pyk2^−/−^ (blue) throughout the experiment. Black arrows indicate the injection point of the treatment; (**B**) Quantification of fluorescence intensity of Fluo4 (left panel) or TMRM (right panel) at basal condition and after TG or FCCP treatment; (**C**) Representative confocal images of Fluo4 (green) and TMRM (red) fluorescence in basal conditions and after TG and FCCP exposure. Scale bar, 10 microns. Data are mean ± SEM. n = 55–62 neurons/genotype from 4 different cultures. * *p* < 0.05, *** *p* < 0.001, **** *p* < 0.0001 vs. basal Pyk2^+/+^; $$$$ *p* < 0.001 vs. Pyk2^+/+^ at each condition determined by two-way ANOVA.

**Figure 5 cells-11-00842-f005:**
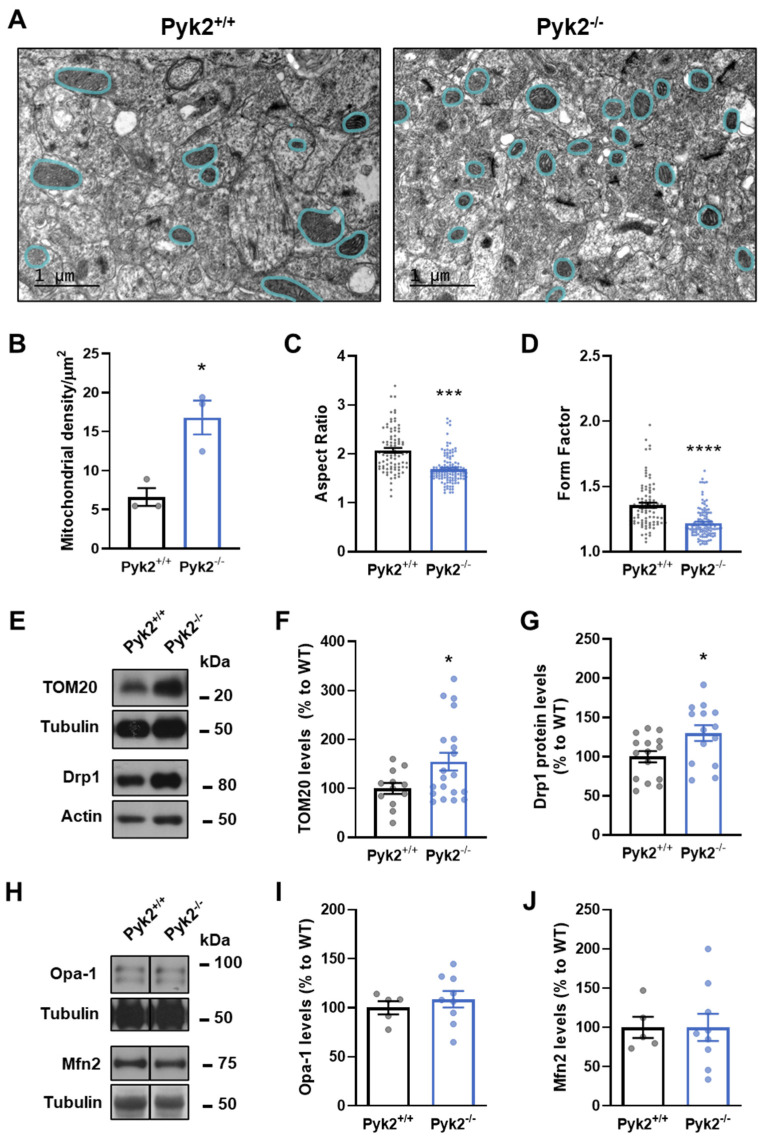
Mitochondrial dynamics and related proteins in hippocampus of Pyk2^−/−^ mice. (**A**) Representative electron microscopy images of pyramidal neurons in hippocampal tissue sections from Pyk2^+/+^ (left panel) and Pyk2^−/−^ (right panel) mice. Blue lines delimit mitochondria. Scale bar, 1 µm; (**B**) Mitochondrial density quantification in Pyk2^+/+^ and Py2^−/−^ hippocampal pyramidal neurons from (**A**); Aspect Ratio (**C**), Mann–Whitney test, (**A**,**B**) = 11,883, 9438, U = 2298; *p* < 0.0001) and Form Factor; (**D**), Mann–Whitney test, A,B = 11,769, 9552, U = 2412; *p* < 0.0001) of results in (**A**) were also determined; (**E**–**G**) Immunoblotting and its quantification of TOM20 ((**E**,**F**), Student *t*-test, *t* = 2.175, df = 30; *p* < 0.05) and Drp1 ((**E**,**G**), Student *t*-test, *t* = 2.455, df = 27; *p* < 0.05) protein levels in dorsal hippocampus of Pyk2^+/+^ and Pyk2^−/−^ mice; (**H**–**J**) Immunoblotting and its quantification of Opa-1 (Student *t*-test, *t* = 0.7007, df = 12; *p* = 0.4968) and Mfn2 (Student *t*-test, *t* = 0.0009984, df = 12; 0.9992) protein levels in hippocampus of Pyk2^+/+^ and Pyk2^−/−^ mice. Actin or tubulin were used as loading control. Molecular weight markers positions are indicated in kDa in (**E**,**H**); Data represent mean ± SEM In (**C**,**D**), n = 87 Pyk2^+/+^ and n = 119 Pyk2^−/−^ mitochondria from 3 different mice per group; Animals, in (**F**), Pyk2^+/+^ n = 12 and Pyk2^−/−^ n = 16; in (**G**), Pyk2^+/+^ n = 15 and Pyk2^−/−^ n = 14; in (**I**,**J**), Pyk2^+/+^ n = 5 and Pyk2^−/−^ n = 9. * *p* < 0.05, *** *p* < 0.001, **** *p* < 0.0001 vs. Pyk2^+/+^.

**Figure 6 cells-11-00842-f006:**
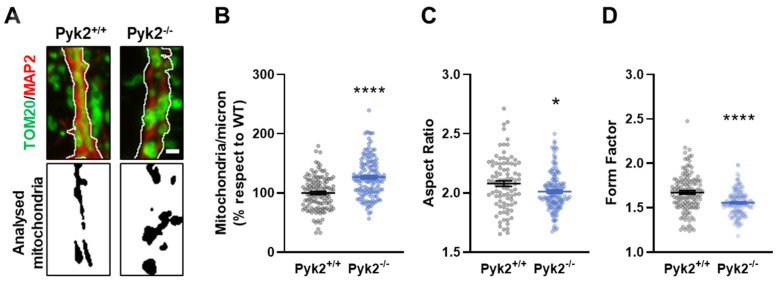
Alterations in mitochondrial morphology in Pyk2^−/−^ primary hippocampal neurons. Hippocampal primary neurons from Pyk2^+/+^ and Pyk2^−/−^ embryos were cultured until DIV21. (**A**) In the upper panel, confocal images of immunolabeling of hippocampal primary neurons with TOM20 (green) and MAP2 (red). White lines delimit the analysed region. In the lower panel, correspondent binary image with the analysed mitochondria in black with white background. Scale bar, 1 micron; (**B**–**D**) Pyk2^−/−^ neurons presented mitochondrial fragmentation as indicated by three different parameters namely: Increased (**B**) number of mitochondria per micron (Student *t*-test, *t* = 6.834, df = 258; *p* < 0.0001); lower (**C**) Aspect Ratio (Mann–Whitney *t*-test, A: 11,659, B: 14906, U: 5036, *p* = 0.0101) and lower (**D**) Form Factor (Mann–Whitney *t*-test, A: 20,567, B: 15,748, U: 5878, *p* < 0.0001). Data are means ± SEM of 15–20 neurons from 8 different embryos per genotype (n = 128–140). * *p* < 0.05, **** *p* < 0.0001 vs. Pyk2^+/+^.

**Figure 7 cells-11-00842-f007:**
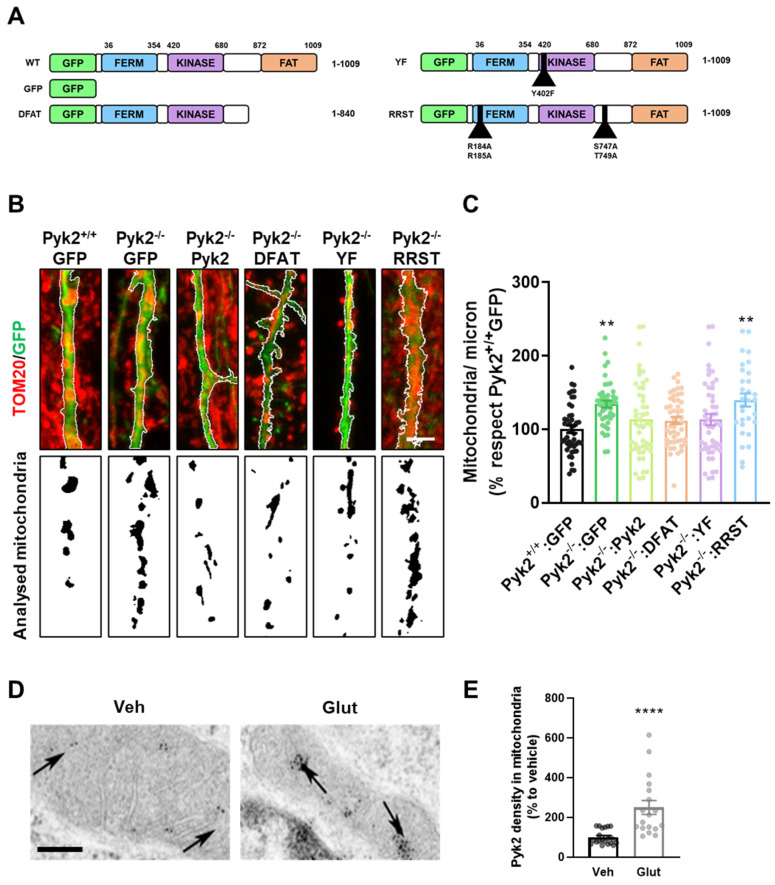
Mechanisms of Pyk2-dependent modulation of mitochondrial density. (**A**) Cultured hippocampal neurons were transfected with plasmids expressing only GFP or Pyk2 or Pyk2 with the indicated mutations namely: DFAT (Pyk2 without the entire FAT domain), YF (Pyk2 with a point mutation in the tyrosine 402 residue for an alanine) and RRST (with four-point mutations in the import/export nuclear domains called R184A, R185A, S747A and T749A); (**B**) Hippocampal primary neurons from Pyk2^+/+^ and Pyk2^−/−^ mice at 21 DIV. In the upper panel, immunostaining of TOM20 (red) and GFP expressed by the plasmids (green). In the lower panel, correspondent binary image with the analysed mitochondria in black with white background. Scale bar, 3 microns; (**C**) Quantification of number of mitochondria as in (**B**) (one-way ANOVA, F_(5, 258)_ = 4.881, *p* = 0.0003). Tukey’s multiple comparisons test was used as a post hoc. Tukey’s *post hoc* p values are: ** *p* < 0.01 vs. Pyk2^+/+^:GFP group; n = 35–40 neurons/group from 4 different cultures; (**D**) Pyk2 presence (gold particles) in the mitochondria of Pyk2^+/+^ cultured hippocampal neurons upon vehicle or glutamate treatment for 15 min (40 µm, electronic microscopy imaging). Scale bar, 0.1 µm; (**E**) Quantification of results (relative to vehicle group) as in (**D**) (Mann–Whitney test, (**A**,**B**) = 271, 259, U = 40; **** *p* < 0.0001). n = 21/20 mitochondria/group from 7 different cultured pyramidal neurons/group. Data are means ± SEM.

## Data Availability

The datasets generated during and/or analysed during the current study are available from the corresponding author on reasonable request.

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
