# Peer review of "Pyk2 Regulates MAMs and Mitochondrial Dynamics in Hippocampal Neurons"

_cells, 2022, doi:10.3390/cells11050842_

Round 1

Reviewer 1 Report

This is a beautiful research paper investigating the role of Pyk2, a non-receptor tyrosine kinase, as a signalling hub in calcium regulation at ER-mitochondrial contact sites in the context of synaptic plasticity.

Their results propose Pyk2 to be relevant in the modulation of ER-mitochondria calcium efflux, affecting mitochondrial function. 

I think, this is a paper of high quality, with a diverse set of well described methods, that underline the statements the authors make. The concept is clear. I only have some minor points:

Figure 1: 

1C: GO Terms labeling too small 

1B: Heatmap legend is missing

Figure 2:

2A: for non-electroon microscopy specialists hard to judge. Here a larger overview needs to be provided with all relevant organelles being clearly indicated, and then a magnification picture with more contrast similar to 3B.

2B: the fractionation blot is not very clear.

Figure 3:

In the legend of Figure 3 the number of ne for 3A and 3B must be more precisely indicated. 

3A: n= from pooled samples? technical replicates???

3B: N= number of animals, sections?

Heading: „3.4. Pyk2 modulates mitochondrial dynamics in vivo and in vitro „ is kind of misleading. I expected trafficking results. Maybe include the morphology aspect? morphological dynamics?

line 52: mitochondria ARE Fundamental regulators

Line 288: There are two-point ".." after (figure 1A).  

Author Response

We thank the reviewers for their thoughtful comments. We have addressed all the questions that required changes in text sections, strengthening the paper, and we indicate its exact new location in our point-by-point reply.

Please note that the modified text appears into the manuscript marked up using the “Track Changes”

  • Figure 1. 1C: GO Terms labelling too small. 1B: Heatmap legend is missing

We have modified labels of GO terms in Figure 1c. Some of the last terms have been removed in order to make the labels bigger and easier to read. This does not suppose a loss of information since all of the results regarding to mass spectrometry are located in supplementary tables 1 and 2. We have added the heatmap legend at the top of Figure 1b.

  • Figure 2. 2A: for non-electron microscopy specialists hard to judge. Here a larger overview needs to be provided with all relevant organelles being clearly indicated, and then a magnification picture with more contrast similar to 3B.

We agree with the reviewer that interpretation of electron microscopy could be clarified. Thus, we have delimited the organelles of importance, showing mitochondria in blue and nucleus in yellow and figure legend has been modified accordingly.

  • 2B: the fractionation blot is not very clear.

Brightness and contrast have been adjusted homogeneously to facilitate interpretation of the fractionation blot in Figure 2b. Row blots are attached as supplementary data.

  • Figure 3. In the legend of Figure 3 the number of ne for 3A and 3B must be more precisely indicated. 3A: n= from pooled samples? technical replicates??? 3B: N= number of animals, sections?

We apologize for the unclear information in the legend of Figure 3. We have specified what represents each sample at lines 364-365 and 371.

Heading: „3.4. Pyk2 modulates mitochondrial dynamics in vivo and in vitro „ is kind of misleading. I expected trafficking results. Maybe include the morphology aspect? morphological dynamics?

We agree with the reviewer that this results heading is not clear. Therefore, we have re-phrased the sentence to “Pyk2 modulates mitochondrial morphology in vivo and in vitro”.

  • Line 52: mitochondria ARE Fundamental regulators

We apologize for this grammar mistake. We have corrected it in the text. We thank to the reviewer for this observation.

  • Line 288: There are two-point ".." after (figure 1A).

We have corrected the typo mistake in the text. We thank to the reviewer for this observation.

Reviewer 2 Report

The article under review suggests novel function for Pyk2 (proline-rich tyrosine kinase 2) in regulation of ER-mitochondrial contacts, mitochondrial morphology and dynamics, and calcium efflux from ER and mitochondria in hippocampal neurons. To draw these conclusions, the authors examine hippocampal neurons in Pyk2+/+ and Pyk2-/- mice using electron microscopy, biochemical and fluorescence-based techniques. While the possible involvement of Pyk2 in interaction between ER and mitochondria is of interest, supporting evidence does not seem to be sufficient:

  1. There is not enough evidence to make conclusions on involvement of Pyk2 in calcium dynamics in ER-mitochondrial contacts. First, to carefully study calcium dynamics in cytosol it is preferrable to use ratiometric probes (such as fura-2) as intensity of non-ratiometric probes can be affected by many factors including changes in pH, etc, which is of particular importance when you use FCCP, which acidifies the cytosol. Second, to draw conclusions on the mitochondrial calcium it is preferrable to use genetically-encoded mitochondrial calcium sensors (such as mtGamP6F, mtRCamP, etc) to measure calcium dynamics in mitochondria during cytoplasmic calcium elevations. Third, it is very unclear how statistics in Figure 4 (B) was calculated. Why all normalizations were done to WT cells? When you look at the images and traces it is very clear that fccp elicit the same response in WT and Pyk2-/- neurons both in fluo-4 and tmre fluorescence.
  2. It is unclear if negative controls were done for Pyk2 staining in electron microscopy. Another question is if electron microscopy data in culture and in vivo match between each other. Authors do not give the quantification for in vivo only for contacts, while in culture they characterize mitochondria shape. It will be good to see if the effects of Pyk2-/- are the same in vivo and in vitro as from exemplary electron microscopy images it seems that they are different. Also method section would benefit if it is described in more details how ER-mitochondrial contacts were quantified. 

I hope this comments will help authors to go further in their study.

Author Response

We thank the reviewers for their thoughtful comments. We have addressed all the questions that required changes in text sections, strengthening the paper, and we indicate its exact new location in our point-by-point reply.

Please note that the modified text appears into the manuscript marked up using the “Track Changes”

Reviewer #2:

  • There is not enough evidence to make conclusions on involvement of Pyk2 in calcium dynamics in ER-mitochondrial contacts. First, to carefully study calcium dynamics in cytosol it is preferrable to use ratiometric probes (such as fura-2) as intensity of non-ratiometric probes can be affected by many factors including changes in pH, etc, which is of particular importance when you use FCCP, which acidifies the cytosol. Second, to draw conclusions on the mitochondrial calcium it is preferrable to use genetically-encoded mitochondrial calcium sensors (such as mtGamP6F, mtRCamP, etc) to measure calcium dynamics in mitochondria during cytoplasmic calcium elevations. Third, it is very unclear how statistics in Figure 4 (B) was calculated. Why all normalizations were done to WT cells? When you look at the images and traces it is very clear that fccp elicit the same response in WT and Pyk2-/- neurons both in fluo-4 and tmre fluorescence.

We agree with the reviewer that nowadays there are more accurate methodology for the study of cytosolic and mitochondrial calcium dynamics. However, we took advantage of the technic that was already settled in our laboratory and validated in our primary neuronal cultures [1]. We followed the protocol published by Michael R. Duchen and colleagues [2], where Fluo4 and TMRM were used simultaneously in live cells and FCCP and thapsigargin were added. Combination of FCCP with non ratiometric dyes as TMRM or TMRE has been previously described in several cell types [3,4].     

We did not attempt to address specifically mitochondrial calcium dynamics but the ER-mitochondrial crosstalk regarding calcium transport. We and others have previously reported the combined treatment with thapsigargin and FCCP to study calcium efflux from ER to mitochondria [1,4,5]. Nonetheless, we are aware about the limitations of this approach and the text has been modified accordingly to avoid misleading to unprecise conclusions at lines 518-520.

In Figure 4 B, values were normalized to basal Pyk2+/+. Otherwise, if values were relativized to Pyk2+/+ in each condition, explanation of results appear misleading. This way, we wanted to emphasise that alterations in TMRM in Pyk2-/- were due to a maintenance in the levels and not because of a hyperpolarization of mitochondria.

We expected that in Pyk2-/- neurons, although slight, the increase in levels of cytosolic calcium after thapsigargin would induce mitochondria to absorb this excess of calcium along with a mitochondrial depolarization [6] but this did not occur. This result has prompt us to propose that absence of Pyk2 might be affecting mitochondrial capacity of Ca2+ uptake. However, we agree with the reviewer that effect of FCCP at longer time seemed to level both Fluo4 and TMRM between genotype and this has been clarified at the text (lines 393-397).

Certainly, further explanation about normalization and statistics of this experiment was needed and thus, it has been added to the manuscript and figure legend in line 408. Statistical analysis has been reviewed and corrected.

  • It is unclear if negative controls were done for Pyk2 staining in electron microscopy. Another question is if electron microscopy data in culture and in vivo match between each other. Authors do not give the quantification for in vivo only for contacts, while in culture they characterize mitochondria shape. It will be good to see if the effects of Pyk2-/- are the same in vivo and in vitro as from exemplary electron microscopy images it seems that they are different. Also method section would benefit if it is described in more details how ER-mitochondrial contacts were quantified.

We agree with the reviewer that some clarifications about the results from electron microscopy would be need it. First, an image from a negative control for Pyk2 staining has been added in Figure 2b. To test method specificity of the immunogold staining procedure, the primary antibody Pyk2 was omitted. Under these conditions, no selective labelling was observed. Following reviewer’s #1 advice, relevant organelles in Figure 2b as mitochondria and nucleus have been outlined in blue and yellow respectively. Figure legend has been updated correspondingly at lines 343-345.

We apologise to the reviewer if there was lack of clarity in the text. At line 352 we stand “To further examine these organelles associations, we next measured ER-mitochondria interactions both in vivo and in vitro neurons of Pyk2+/+ and Pyk2-/- mice” and at line 405 “Pyk2 modulates mitochondrial morphology in vivo and in vitro”. However, we have modified the text (line 409) and the figure legends in Figure 2 (lines 365 and 367) and Figure 5 (line 420) for a better understanding if the analysis was performed in in vivo or in vitro model.

Although we did not perform neuronal cultures of Pyk2+/+ and Pyk2-/- for electron microscopy observation, we believe that quantification of mitochondrial morphology and ER-mitochondria contacts sites both in vivo and in vitro were evaluated appropriately using previously validated techniques [7,8], that encourage us to assume that our results are robust enough.

With regard to ER-mitochondria analysis, while electron microscopy is considered as a gold methodology, proximity ligation assay detecting VDAC1-IP3R3 complex has been highly recommended to quantify contacts in vitro microscopy [9–11].

Concerning methodology, a wider explanation about the manual quantification of ER-mitochondria contacts has been added in the “Materials and Methods” section (lines 166-169) and in the legend of Figure 3b (line 371).

References

  1. Cherubini, M.; Lopez-Molina, L.; Gines, S. Mitochondrial fission in Huntington’s disease mouse striatum disrupts ER-mitochondria contacts leading to disturbances in Ca2+ efflux and Reactive Oxygen Species (ROS) homeostasis. Neurobiol. Dis. 2020, 136, 104741, doi:10.1016/j.nbd.2020.104741.
  2. McKenzie, M.; Lim, S.C.; Duchen, M.R. Simultaneous measurement of mitochondrial calcium and mitochondrial membrane potential in live cells by fluorescent microscopy. J. Vis. Exp. 2017, 2017, 2–7, doi:10.3791/55166.
  3. Joshi, D.C.; Bakowska, J.C. Determination of mitochondrial membrane potential and reactive oxygen species in live rat cortical neurons. J. Vis. Exp. 2011, 2–5, doi:10.3791/2704.
  4. Duchen, M.R.; Leyssens, A.; Crompton, M. Transient mitochondrial depolarizations reflect focal sarcoplasmic reticular calcium release in single rat cardiomyocytes. J. Cell Biol. 1998, 142, 975–988, doi:10.1083/jcb.142.4.975.
  5. Eaddy, A.C.; Schnellmann, R.G. Visualization and quantification of endoplasmic reticulum Ca2+ in renal cells using confocal microscopy and Fluo5F. Biochem Biophys Res Commun 2011, 404, 424–427, doi:10.1016/j.bbrc.2010.11.137.Visualization.
  6. Boitier, E.; Rea, R.; Duchen, M.R. Mitochondria exert a negative feedback on the propagation of intracellular Ca2+ waves in rat cortical astrocytes. J. Cell Biol. 1999, 145, 795–808, doi:10.1083/jcb.145.4.795.
  7. Koopman, W.J.H.; Visch, H.J.; Smeitink, J.A.M.; Willems, P.H.G.M. Simultaneous quantitative measurement and automated analysis of mitochondrial morphology, mass, potential, and motility in living human skin fibroblasts. Cytom. Part A 2006, 69, 1–12, doi:10.1002/cyto.a.20198.
  8. Tubbs, E.; Theurey, P.; Vial, G.; Bendridi, N.; Bravard, A.; Chauvin, M.A.; Ji-Cao, J.; Zoulim, F.; Bartosch, B.; Ovize, M.; et al. Mitochondria-associated endoplasmic reticulum membrane (MAM) integrity is required for insulin signaling and is implicated in hepatic insulin resistance. Diabetes 2014, 63, 3279–3294, doi:10.2337/db13-1751.
  9. López-Crisosto, C.; Bravo-Sagua, R.; Rodriguez-Peña, M.; Mera, C.; Castro, P.F.; Quest, A.F.G.; Rothermel, B.A.; Cifuentes, M.; Lavandero, S. ER-to-mitochondria miscommunication and metabolic diseases. Biochim. Biophys. Acta - Mol. Basis Dis. 2015, 1852, 2096–2105.
  10. Benhammouda, S.; Vishwakarma, A.; Gatti, P.; Germain, M. Mitochondria Endoplasmic Reticulum Contact Sites (MERCs): Proximity Ligation Assay as a Tool to Study Organelle Interaction. Front. Cell Dev. Biol. 2021, 9, 1–6, doi:10.3389/fcell.2021.789959.
  11. Yang, M.; Li, C.; Yang, S.; Xiao, Y.; Xiong, X.; Chen, W.; Zhao, H.; Zhang, Q.; Han, Y.; Sun, L. Mitochondria-Associated ER Membranes – The Origin Site of Autophagy. Front. Cell Dev. Biol. 2020, 8, 1–11, doi:10.3389/fcell.2020.00595.

Round 2

Reviewer 2 Report

The authors provide clarifications to their methods and some arguments, but still some concerns remain.

1. Conclusions on the calcium transport between ER and mitochondria do not have experimental proofs. And either more experimental work is required or conclusions have to be modified. Please, see below.

We did not attempt to address specifically mitochondrial calcium dynamics but the ER-mitochondrial crosstalk regarding calcium transport. We and others have previously reported the combined treatment with thapsigargin and FCCP to study calcium efflux from ER to mitochondria [1,4,5].

1) Cited papers indeed examine calcium uptake by mitochondria following calcium release from ER by measuring mitochondrial depolarization utilizing tmre, but this approach cannot be used to study calcium dynamics between these organelles using thapsigargin as it inhibits SERCA ATPase and induce global cytosolic calcium elevation by depleting ER (1). In [4], this drug was not applied, and in [5] ER content was studied without implication to study ER-mito transfer in contacts. Calcium transport between ER and mitochondria is believed to be mediated by IP3 receptors and to study local calcium dynamics in MAMs appropriate methods have to be used (2-4).

The experiments performed by the authors demonstrate that ER depletion in the absence of pyk2 leads to smaller cytoplasmic calcium elevation, which could be a sign of altered calcium capacity of ER in this condition.

In Figure 4 B, values were normalized to basal Pyk2+/+. Otherwise, if values were relativized to Pyk2+/+ in each condition, explanation of results appear misleading. This way, we wanted to emphasise that alterations in TMRM in Pyk2-/- were due to a maintenance in the levels and not because of a hyperpolarization of mitochondria.

Fluo-4 and tmre brightness can be affected by numerous factors and cross-normalization does not seem to be feasible and reasonable (5-7). Such comparison could have been made if ratiometric probes were used (initial ratios in this case could have been compared).

We expected that in Pyk2-/- neurons, although slight, the increase in levels of cytosolic calcium after thapsigargin would induce mitochondria to absorb this excess of calcium along with a mitochondrial depolarization [6] but this did not occur. This result has prompt us to propose that absence of Pyk2 might be affecting mitochondrial capacity of Ca2+ uptake.

Calcium transport into mitochondria strongly depends on the calcium concentration in the cytoplasm or ER/mito contacts (8-9). Since thapsigargin evoked significantly lower cytoplasmic calcium response in the absence of Pyk2, smaller depolarization and smaller calcium are to be expected. Mitochondrial depolarization and hence “calcium absorption” seem to directly correspond to the cytosolic calcium elevation and be similar between the conditions.

2. First, an image from a negative control for Pyk2 staining has been added in Figure 2b. 

Please, show images at the same magnification. I understand that the smaller magnification is to show greater area, but for this purpose several field should be shown perhaps.

Since Fig. 2 and Fig. 5 demonstrate electron images from the same prep, it would be good to mark mitochondria and ER on both of them to demonstrate in these images are consistent with each other.

I hope my comments and suggestion will help the authors to improve their manuscript.

1 Rogers et al., Biosci Rep (1995) 15 (5): 341–349.

  1. Rizzuto et al., Science. 1998 Jun 12;280(5370):1763-6. doi: 10.1126/science.280.5370.1763.
  2. Bartok et al., Nat Commun. 2019 Aug 19;10(1):3726. doi: 10.1038/s41467-019-11646-3.
  3. Booth et al. Redox nanodomains are induced by and control calcium signaling at the ER-mitochondrial interface. Mol. Cell. 2016;63:240–248. doi: 10.1016/j.molcel.2016.05.040.
  4. Kao et al., Methods Cell Biol. 2010;99:113-52. doi: 10.1016/B978-0-12-374841-6.00005-0.
  5. Thomas et al., Cell Calcium. 2000 Oct;28(4):213-23. doi: 10.1054/ceca.2000.0152.
  6. Scaduto et al., Biophys J. 1999 Jan;76(1 Pt 1):469-77. doi: 10.1016/S0006-3495(99)77214-0.
  7. Williams et al., PNAS June 25, 2013 110 (26) 10479-10486; https://doi.org/10.1073/pnas.1300410110
  8. Szabadkai et al., J Biol Chem 2003 Apr 25;278(17):15153-61. doi: 10.1074/jbc.M300180200. Epub 2003 Feb 1

Author Response

We would like to re-submit a revised version of our manuscript cells-1568384 by López-Molina et al. for re-consideration after major revisions at Cells. We thank the reviewer for his/her thoughtful comments. We have addressed all the questions that required changes in text sections changing in turn, some of the interpretations and conclusions. We honestly thing that these changes give more strength to the manuscript, and we indicate their exact new location in our point-by-point reply.

Please note that the modified text appears into the manuscript marked up in red

Reviewer #2:

Cited papers indeed examine calcium uptake by mitochondria following calcium release from ER by measuring mitochondrial depolarization utilizing tmre, but this approach cannot be used to study calcium dynamics between these organelles using thapsigargin as it inhibits SERCA ATPase and induce global cytosolic calcium elevation by depleting ER (1). In [4], this drug was not applied, and in [5] ER content was studied without implication to study ER-mito transfer in contacts. Calcium transport between ER and mitochondria is believed to be mediated by IP3 receptors and to study local calcium dynamics in MAMs appropriate methods have to be used (2-4).

The experiments performed by the authors demonstrate that ER depletion in the absence of pyk2 leads to smaller cytoplasmic calcium elevation, which could be a sign of altered calcium capacity of ER in this condition.

Considering that updated literature provides more accurate methodology to monitor mitochondria calcium, we agree that our approach can only address conclusions about calcium homeostasis regarding intracellular calcium. Hence, expressions as “ER-mitochondrial calcium transfer” has been replaced for “calcium homeostasis”. We thank the reviewer for this clarification and this concept has been added to the text at

Abstract: Line 27.

Methods section: Lines 236.

Results section: The entire section 3.3 (lines 371-393), including the title of the section, regarding to the calcium experiments. The text in figure legend 4 has been also changed accordingly.

Discussion section: a wider explanation has been included in the discussion at lines 508-518.

Fluo-4 and tmre brightness can be affected by numerous factors and cross-normalization does not seem to be feasible and reasonable (5-7). Such comparison could have been made if ratiometric probes were used (initial ratios in this case could have been compared).

Values expressed as fold-changes have been replaced for the absolute values of fluorescence intensity both for Fluo4 and TMRM. Statistical analysis has been re-evaluated (Figure legend 4 and lines 380-393) and section of materials and methods have been corrected (lines 247-248).  

Calcium transport into mitochondria strongly depends on the calcium concentration in the cytoplasm or ER/mito contacts (8-9). Since thapsigargin evoked significantly lower cytoplasmic calcium response in the absence of Pyk2, smaller depolarization and smaller calcium are to be expected. Mitochondrial depolarization and hence “calcium absorption” seem to directly correspond to the cytosolic calcium elevation and be similar between the conditions.

We have rephrased the results sections at lines 393-398 according to the reviewer’ suggestions (The entire section 3.3; lines 371-393). We thank the reviewer for this clarification and improvement of the understanding of this experiment. 

Please, show images at the same magnification. I understand that the smaller magnification is to show greater area, but for this purpose several field should be shown perhaps.

We agree with the reviewer that electron microscopy images from negative control could be improved. Similar magnifications to Figure 2a showing mitochondria, nucleus and MAMs have been added with the corresponding scale and modifications in the figure legend.

Since Fig. 2 and Fig. 5 demonstrate electron images from the same prep, it would be good to mark mitochondria and ER on both of them to demonstrate in these images are consistent with each other.

We thank the reviewer for this observation. We have delimited mitochondria in blue and ER in red in bigger magnification images of electron microscopy in Figure 2a and 2b. Figure legend has been modified accordingly. Additionally, mitochondria have also been outlined at Figure 5a with blue lines.